# Detecting m⁶A at single-molecular resolution via direct RNA sequencing and realistic training data

Adrian Chan[1,4], Isabel S. Naarmann-de Vries[1,2,4], Carolin P. M. Scheitl[3,4], Claudia Höbartner ●[3] & Christoph Dieterich ●[1,2] ✉

Direct RNA sequencing offers the possibility to simultaneously identify canonical bases and epi-transcriptomic modifications in each single RNA molecule. Thus far, the development of computational methods has been hampered by the lack of biologically realistic training data that carries modification labels at molecular resolution. Here, we report on the synthesis of such samples and the development of a bespoke algorithm, mAFiA (m⁶A Finding Algorithm), that accurately detects single m⁶A nucleotides in both synthetic RNAs and natural mRNA on single read level. Our approach uncovers distinct modification patterns in single molecules that would appear identical at the ensemble level. Compared to existing methods, mAFiA also demonstrates improved accuracy in measuring site-level m⁶A stoichiometry in biological samples.

Each time-trace in direct RNA-sequencing (dRNA-Seq) on the Oxford Nanopore platform encodes the unique modification fingerprint of an individual molecule[1]. However, no database of naturalistic RNA with modification labels at the read-level exists yet. For m⁶A specifically, methods development thus far is based on either synthetic molecules with no biological resemblance[2], or natural mRNA with approximate labels[3]. In this work, we bridge this gap by synthesizing short RNA oligos that recreate sections of actual mRNA with m⁶A sites, while the modification status of each nucleotide (nt) is precisely controlled.

## Results and discussion

Our synthetic samples cover the six most common DRACH motifs, which collectively account for almost 80% of all consensus m⁶A sites in human mRNA[4–6]. For each motif, at least two different sequence designs are chosen from the human transcriptome with known m⁶A loci, in the form of 21- or 33-mers (Supplementary Table 1). Altogether, our training dataset consists of 15 sequences, each as an unmodified (UNM) or modified (MOD, m⁶A inserted in the sequence center) variant. To produce polymers suitable for dRNA-Seq, the oligos are ligated into longer RNA molecules using two different enzymatic approaches, either by random ligation (RL) or splint ligation (SL) (Fig. 1a, Methods, Supplementary Figs. 1, 2).

To utilize such synthetic data, we developed a simple method whose model complexity is low compared to the available sample sizes. Our m⁶A Finding Algorithm (mAFiA) makes use of hidden features generated by a fully-convolutional basecaller RODAN[7]. Instead of predicting the sequence itself, mAFiA uses the extracted features to assign an m⁶A probability, $P(m^6A)$, to a specific nucleotide on a single read (Fig. 1b, Methods). Our approach does not interfere with the accuracy of the original basecaller. Moreover, the ability to detect new modification patterns, including non-m⁶A ones, can be readily extended as training data becomes available, without the need to retrain the entire set of models from scratch.

We cross-validated our method on the RL and SL datasets, which involve different sequence designs and are synthesized through different ligation strategies in two independent laboratories. Each dataset is split into 75% for training and 25% for validation. Figure 1c shows the predictions of mAFiA trained on the RL samples and validated on SL. For each DRACH motif, there is a stark divergence in $P(m^6A)$ assigned to single nucleotides on each UNM or MOD read. The precision-recall curves on all motifs (Fig. 1d) suggest that our model trained on one dataset performs robustly in hitherto unseen sequence contexts (Supplementary Fig. 3a). We repeat the cross-validation using different combinations of train and validation datasets, and evaluate the

[1]Klaus Tschira Institute for Integrative Computational Cardiology, University of Heidelberg, Heidelberg, Germany. [2]German Centre for Cardiovascular Research (DZHK)-Partner Site Heidelberg/Mannheim, Heidelberg, Germany. [3]Institute of Organic Chemistry, University of Würzburg, Würzburg, Germany. [4]These authors contributed equally: Adrian Chan, Isabel S. Naarmann-de Vries, Carolin P. M. Scheitl. ✉e-mail: christoph.dieterich@uni-heidelberg.de

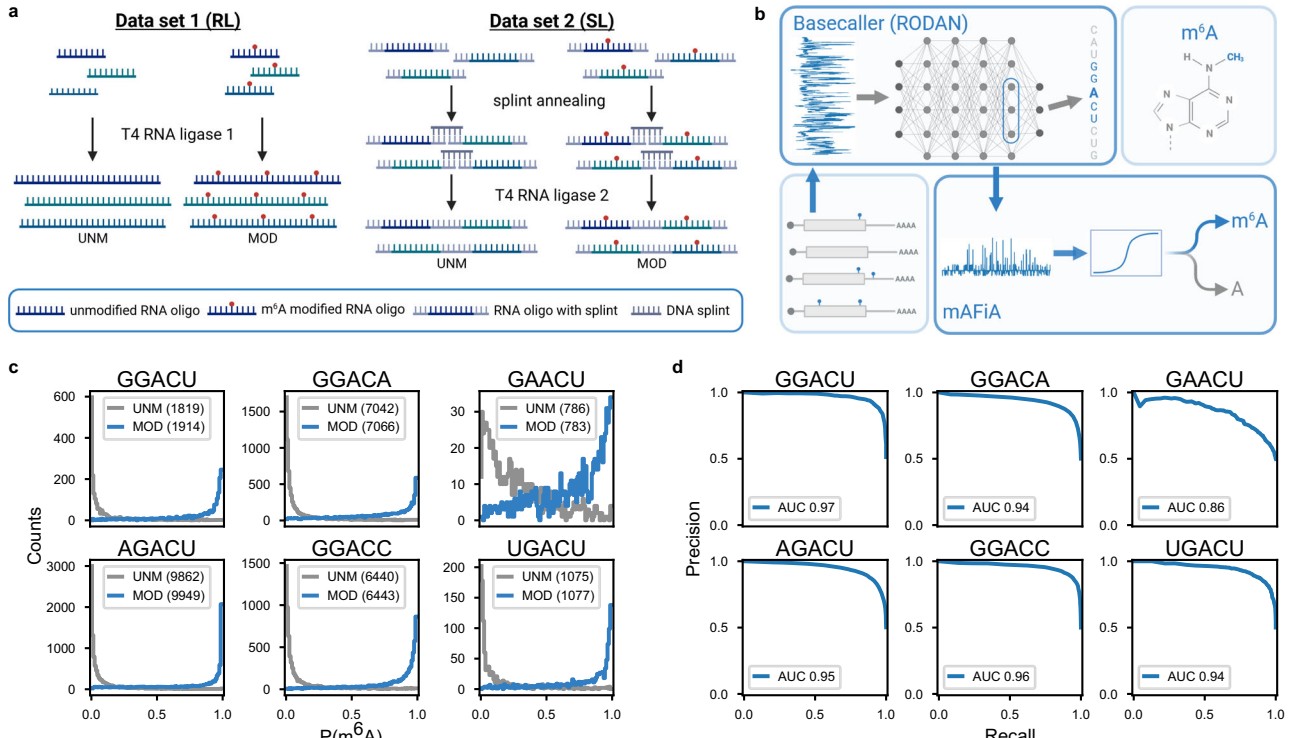

**Fig. 1 | Training on synthetic RNA. a** Training data - Schematic representation of the two different ligation strategies - random ligation (RL) and splint ligation (SL). Left: 21 nt RNA oligos (colors indicate different sequences) with a central DRACH motif (A or m⁶A (red dot)) are concatenated to homopolymers by RNA ligase 1 and sequenced in unmodified (UNM) or modified (MOD) pools. Right: 33 nt RNA oligos with a central DRACH motif (A or m⁶A (red dot)) and flanking splint sequences (grey) are ligated to heteropolymers by splint-assisted ligation using RNA ligase 2 and sequenced in UNM or MOD pools. **b** Algorithm - From the backbone basecaller network RODAN[7], mAFiA extracts a 768-dimensional feature vector **x** that corresponds to a predicted nucleotide A in one of the target motifs. Logistic regression

(see methods) is then applied to **x** to generate the read-level m⁶A modification probability, $P(m^6A)$, where $0 \leq P(m^6A) \leq 1$. **c** Model validation - mAFiA models trained on 75% of the RL dataset and validated on 25% of SL. The histogram shows the distribution of predicted $P(m^6A)$ assigned to central A nucleotides mapped to various motifs, in unmodified (UNM) and modified (MOD) replicates. Gray lines correspond to nucleotides from the UNM sample and blue lines from MOD. Numbers in brackets are the validated sample size. **d** Precision-recall curves (PRCs) calculated from the $P(m^6A)$ distributions in (**c**), with area-under-curve (AUC) given in the legend. Source data are provided as a Source Data file.

model's performance using the average AUC of all 6 motifs. Not surprisingly, the model resulting from a joint dataset performs better than those trained on either sample alone (Supplementary Fig. 3b). Thereafter, all evaluations are performed using the combined model.

To establish a set of single-molecule benchmarks, we designed two experiments in which each strand of RNA contains both MOD and UMD oligos in unknown order. TEST1 involves heteropolymers that contain either the oligo combination {UGm⁶ACU…GGACC}, or {UGACU…GGm⁶ACC} in one molecule (Fig. 2a, top). The $P(m^6A)$ predicted by mAFiA in each sample shows that it can indeed distinguish the motifs on each read where modification occurs (Fig. 2b). We subjected TEST1 to two other methods that aim to detect m⁶A at single-molecule resolution (CHEUI and m6Anet)[2,3], and observe that all three approaches perform robustly (Fig. 2c).

TEST2 involves RNA strands which contain periodic repetition of an identical 13-mer sequence, with GGACU in the center (Fig. 2a, bottom). The sample contains two variants, in which every RNA strand can have periodic m⁶A insertion either at positions {7,33,59,…}, or {20,46,72,…}. In other words, each dRNA read contains m⁶A signals at intervals of 26 nts, but this pattern can be shifted from one read to another. On the aggregate level, all the GGACU locations are indistinguishable and show modification ratios of ~50% (Fig. 2d, bottom). However, mAFiA is able to distinguish the underlying m⁶A patterns in each read (Fig. 2d, top). Denoting the positions at {0,26,52,…}nts from the highest $P(m^6A)$ position on a read as "even," and those at {13,39,65,…}nts as "odd," mAFiA detects a sharp contrast in $P(m^6A)$ between these two sets of positions (Fig. 2e). Analogous calculations

with CHEUI and m6Anet show considerable deterioration in their performance (Fig. 2f).

Having validated our method on synthetic molecules, we then apply it to human mRNA from HEK293 cells. Figure 3a shows the alignment of dRNA reads covering the entire transcript of *HSPA1A*, a highly expressed gene in HEK293. The green shading shows the predictions of mAFiA on different transcript positions (alignment columns) of individual reads (rows). Neighboring sites in close proximity can be seen to exhibit highly varying distributions of $P(m^6A)$ (Fig. 3b). To aggregate the distribution of read-level $P(m^6A)$ on an alignment column into site-level modification ratio, we define the site stoichiometry $S$ as the fraction of modified nucleotides, i.e., nts with $P(m^6A) \geq 0.5$, aligned to that site (see also definition of $S$ in Methods). Using this site-level metric, we compare the transcriptome-wide m⁶A profile between HEK293 wild type (WT) and METTL3 knock out (KO) samples (Supplementary Fig. 4h). While the chromosomal locations of methylation sites are conserved between the two samples, systematic down-regulation can be observed in the KO dataset. Across all chromosomes, a site-by-site comparison of $S$ indicates that most of the down-regulation occurs at locations that are highly methylated in WT (Fig. 3c).

To assess the quantitative accuracy of mAFiA's site-level predictions, we compare $S_{mAFiA}$ against the modification stoichiometry published by GLORI[8], $S_{GLORI}$, which ranges from 10% to 100%. Figure 3d shows good agreement between the two modalities among all 6 target motifs. Across 5925 sites on the human transcriptome, $S_{mAFiA}$ exhibits a correlation of 0.86 with $S_{mAFiA}$. On the same set of sites, the predictions

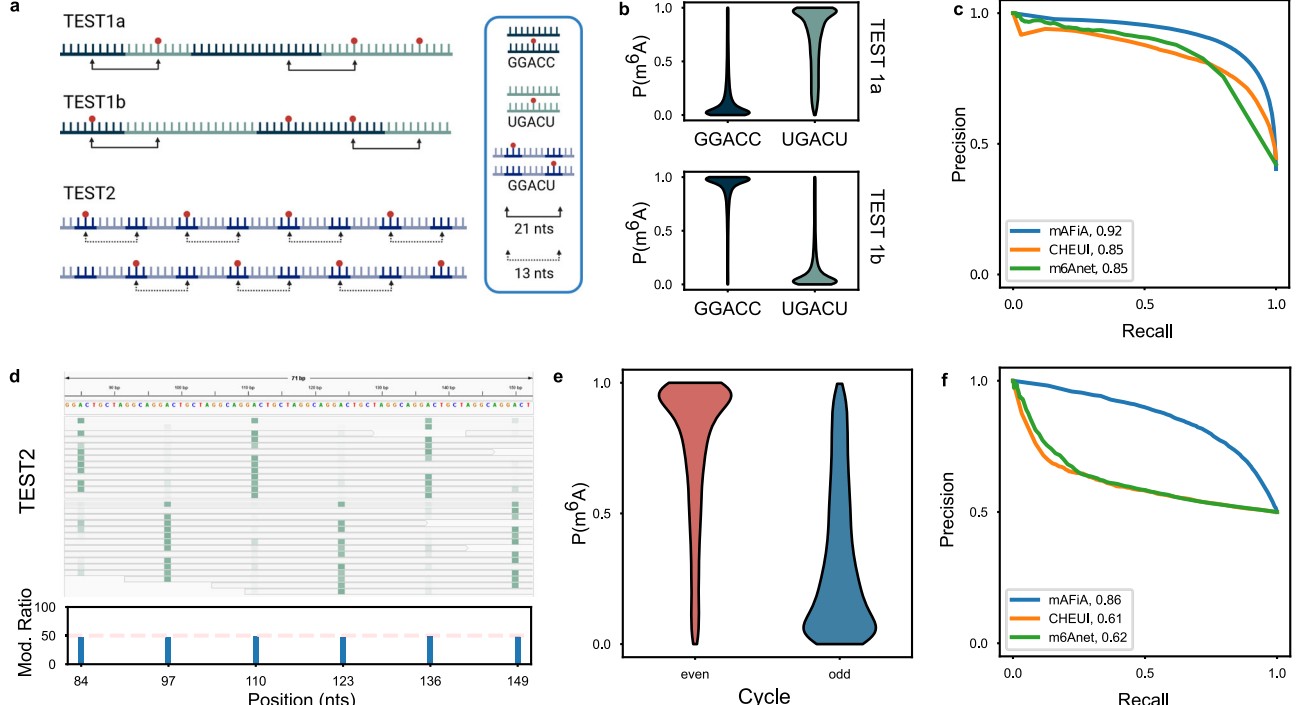

**Fig. 2 | Validation on synthetic RNA. a** Test data - Top: TEST1 consists of RNA heteropolymers of two types of oligos, one modified and one unmodified, that are randomly ligated. Red dot signifies the insertion of an m⁶A. TEST1a contains the motifs UGm⁶ACU and GGACC, while Test1b contains UGACU and GGm⁶ACC. Bottom: TEST2 consists of oligos with the GGACU motif in alternating unmodified and modified forms. The sequence repeats every 13 nts, while m⁶A is present in 26 nt-cycles. The underlying m⁶A pattern can be shifted by 13 nts from one molecule to another. **b** TEST1 - Violin plots showing the distribution of $P(m^6A)$ assigned to A nucleotides mapped to GGACC (dark blue) or UGACU (green) in TEST1a (top) and TEST1b (bottom). The $P(m^6A)$ distribution in TEST1a shows that most of the A nucleotides in UGACU have high modification probability, whereas those in GGACC have $P(m^6A)$ close to 0. The $P(m^6A)$ contrast between the two motifs is flipped in TEST1b. Sample sizes: N(TEST1a | GGACC) = 42280, N(TEST1a | UGACU) = 20849, N(TEST1b | GGACC) = 18854, N(TEST1b | UGACU) = 16569. **c** PRC calculated from the $P(m^6A)$ distribution in (**b**), where m⁶A detections aligned to UGACU in TEST1a and

GGACC in TEST1b count as true-positives (mAFiA in blue). Results for CHEUI (orange) and m6Anet (green) are also included. Numbers in the legend refer to the respective AUCs. **d** TEST2 - (top) IGV snapshot of mAFiA results in TEST2, with green shading proportional to $P(m^6A)$ assigned to single nucleotide positions on single reads. In each read, signal peaks occur at periods of 26 nts, separated by troughs with low $P(m^6A)$. Two populations with a phase-shift of 13 nts are clearly distinguished. (bottom) Aggregate modification ratios at the site-level, where all GGACU sites are indistinguishable. **e** Violin plot showing the distribution of $P(m^6A)$ assigned to A nucleotides mapped to "even" (red) or "odd" (blue) GGACU cycles. "Even" positions are defined as those at distances of $(2n \times 13)nts$ from the nucleotide with the highest $P(m^6A)$ assigned on a read, while "odd" ones occur at distances $(2n+1) \times 13nts$, where $n$ is any integer. Sample size N(GGACU) = 13648. **f** PRC calculated from the $P(m^6A)$ distribution in (**e**), where detections in the even cycles are counted as true-positive. Numbers in legend are AUCs produced by each method. Source data are provided as a Source Data file.

of CHEUI and m6Anet yield correlations of 0.64 and 0.80 respectively (Supplementary Fig. 4e,f). In both synthetic RNA and HEK293 benchmarks, mAFiA outperforms existing single-molecule methods (Supplementary Fig. 4g).

In addition to wildtype (WT) samples, we also prepared mixtures of HEK293 WT with in vitro transcribed (IVT) RNA at different concentrations. HEK293 IVT contains identical RNA sequences as in WT, but is devoid of all modifications. As the WT sample is diluted to a fraction $f_{WT}$ of its original concentration, we expect the measured stoichiometry of a site to be reduced to $f_{WT} \times S_{orig}$, where $S_{orig}$ is the original modification ratio of the site in full concentration. Figure 3e resoundingly confirms the stoichiometric sensitivity of mAFiA, where, across thousands of sites on the transcriptome, it correctly observes the system-wide reduction of site-level modification ratios, with an observed coefficient $m(\frac{S_{mAFiA}}{S_{GLORI}})$ that agrees with various $f_{WT}$ (Fig. 3f).

While mAFiA is optimized for the most common m⁶A patterns in the human transcriptome, we evaluated its applicability also in a wider context. Testing on samples of *Arabidopsis thaliana* dRNA data shows good correspondence to previously published miCLIP measurements[9] (Fig. 3g). Out of 522 high-modification sites predicted (sites with $S_{mAFiA} \geq 50\%$), 372 (71%) coincide with a miCLIP peak within 5 nts. The agreement rises to 82% if we consider only the more confident sites with $S_{mAFiA} \geq 80\%$. A site-by-site comparison of the predicted m⁶A

profiles between the *col0* (wildtype) and *vir1* (mutant) strains shows a transcriptome-wide down-regulation in the otherwise highly modified sites (Fig. 3h and Supplementary Fig. 6). We note that the overall coverage of m⁶A sites in a specific species can be further improved with bespoke training data, although the primary use case of mAFiA remains mammalian RNAs.

Through meticulous validation in both synthetic and natural RNAs, we have demonstrated an accurate, proof-of-principle m⁶A finding algorithm at single-molecule resolution, which also allows detection of isoform modification stoichiometry (Supplementary Fig. 5). As new training data becomes available in the future, the method can be expanded to the full set of DRACH motifs and additional RNA modifications.

## Methods
### RNA oligos
Synthetic RNA oligos with a 5′ phosphate were purchased or prepared in house by in vitro transcription or solid-phase synthesis using 2′-O-TOM-protected RNA phosphoramidite as previously described[10] and outlined below. Purchased oligos for random ligation (GenScript, purified by RNase free HPLC) were of 21 nts length; in house prepared oligos for splinted ligation were 33 nts in length. Each oligo was synthesized either as unmodified oligo (UNM) or with an m⁶A at the

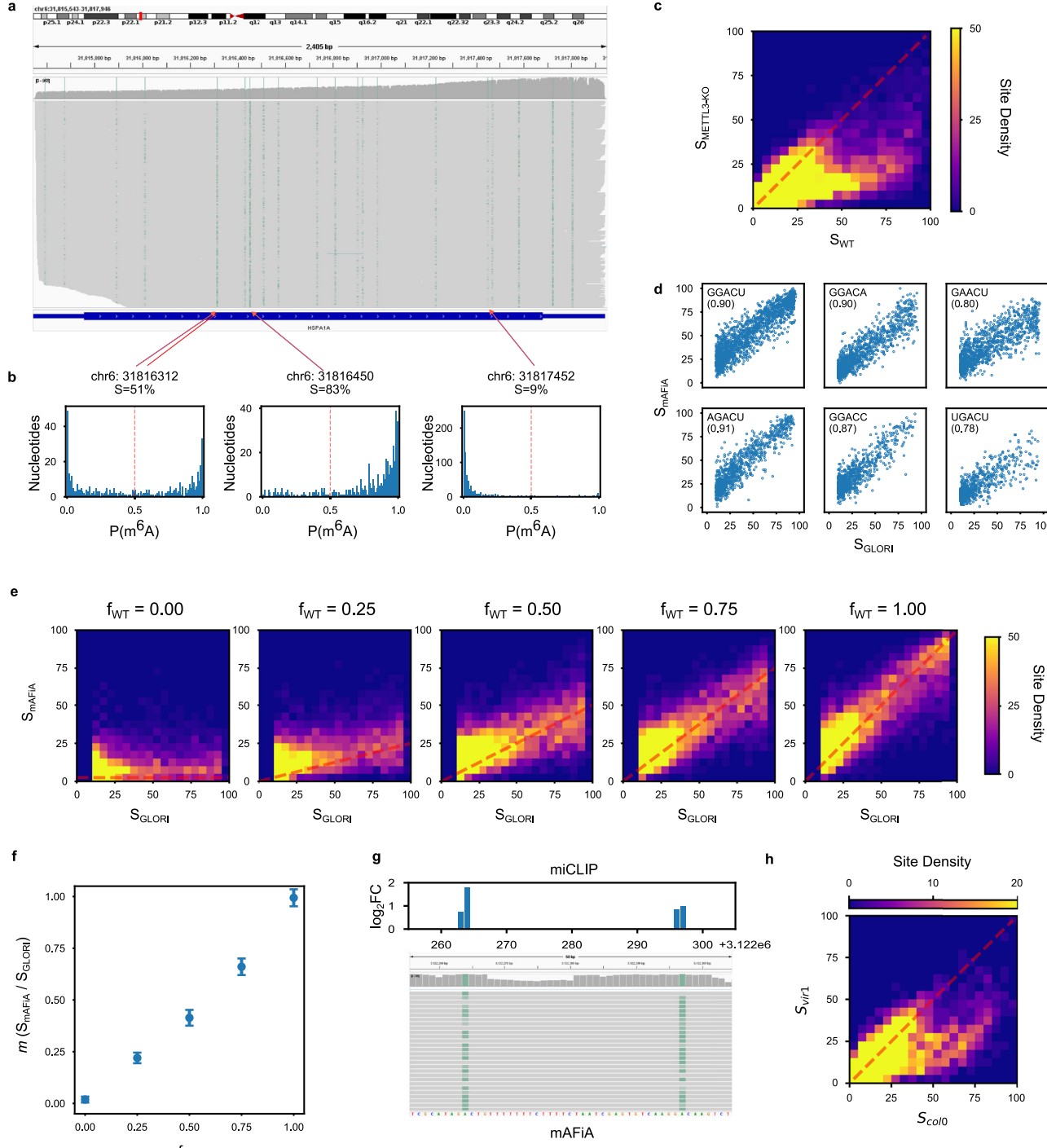

**Fig. 3 | Testing on biological mRNA. a** Full-transcript m⁶A sites - IGV snapshot of the *HSPA1A* gene in HEK293 cells, green shades indicate the read-level modification probability, $P(m^6A)$, of a single nucleotide on each read. **b** From read to site level - $P(m^6A)$ distributions of single nucleotides at 3 locations along *HSPA1A*, showing balanced (left), highly (center), and lowly (right) modified sites. The site-level stoichiometry, $S$, is defined as the fraction of nucleotides with $P(m^6A) \geq 0.5$ at that site (see Methods). **c** Whole-transcriptome m⁶A profile - Site-by-site comparison of $S$ predicted by mAFiA in HEK293 WT versus METTL3-KO, across all chromosomes. Site-density represents the number of sites within each 5% bin. n = 15316 sites. The red dashes mark the bins where $S_{WT} = S_{KO}$. **d** Comparison with GLORI - Scatter plot of site-level m⁶A stoichiometry predicted by mAFiA ($S_{mAFiA}$, y-axis) versus values published by GLORI ($S_{GLORI}$, x-axis), across all chromosomes of HEK293 WT. Numbers in brackets are correlations between ($S_{mAFiA}$, $S_{GLORI}$). GLORI does not report values of $S$ below 10%. n = 5925 sites. **e** Titration experiment – 2d density-plots of site-level stoichiometry comparison between $S_{mAFiA}$ (y-axis) and $S_{GLORI}$ (x-axis), in 5 mixtures of HEK293 WT and IVT (WT fraction $f_{WT}$ = 0.00, 0.25, 0.50, 0.75, 1.00). Red dashes correspond to the expected distribution of site-level stoichiometry depending on WT fraction: $f_{WT} \times S_{GLORI}$. **f** Slope extracted from linear regression of $S_{mAFiA}$ against $S_{GLORI}$, $m\left(\frac{S_{mAFiA}}{S_{GLORI}}\right)$ (y-axis), in (**e**), as a function of $f_{WT}$ (x-axis). The observed variable, $m\left(\frac{S_{mAFiA}}{S_{GLORI}}\right)$, as measured by mAFiA through the system-wide distribution of individual site stoichiometries, largely agrees with the underlying control variable $f_{WT}$. n = {2515, 3903, 6192, 5801, 5925} sites. Data are presented as fitted values +/− standard error. **g** Application to non-mammalian species - IGV snapshot of m⁶A sites detected by mAFiA (bottom) in *Arabidopsis thaliana*, juxtaposed with miCLIP peaks[9] (top). **h** Site-by-site comparison of mAFiA-predicted site-level stoichiometries in wild type *col0* ($S_{col0}$, x-axis) and mutant *vir1* ($S_{vir1}$, y-axis) strains of *Arabidopsis thaliana*. The mutant strain shows significant down-regulation of m⁶A levels in otherwise highly modified sites. N = 11881 sites. Red dashes mark the bins where $S_{col0} = S_{vir1}$. Source data are provided as a Source Data file.

central position (MOD). The oligos cover the six most abundant DRACH motifs according to GLORI-seq. The 5′ and 3′ context was chosen based on ligation efficiency as well as the edit distances between them that allows one to distinguish different oligo fragments in a given sequencing pool. Purchased RNA oligos (Supplementary Table 1) were dissolved at 100 μM in nuclease-free water and stored at −80 °C.

## Solid phase RNA synthesis

Short RNA oligonucleotides modified with m⁶A were synthesized in house using solid phase synthesis on a K&A DNA/RNA-Synthesizer (H6/H-8) with standard phosphoramidite chemistry (2′-O-TOM protected) on controlled pore glass solid support[11]. Phosphoramidites for incorporation of unmodified nucleosides were purchased from ChemGenes (No. ANP-3201, ANP-3202, ANP-3203, ANP-3205). Phosphoramidites for m⁶A incorporation were prepared in house following published procedures[10]. The terminal 5′ phosphate to enable RNA ligation was introduced using 5′-O-DMT-2,2′-sulfonyldiethanol synthesized according to a published procedure[12]. Deprotection of the RNA oligonucleotides was performed in two steps using a mixture of ammonia and methylamine in aqueous solution (1:1) for 6 h at 37 °C, followed by treatment with 1 M tetrabutylamoonium fluoride (TBAF) in THF at 25 °C for 14–16 h. Deprotected oligonucleotides were first desalted and then purified via denaturing PAGE followed by extraction and precipitation with Ethanol.

## In vitro transcription of unmodified RNA oligos

Unmodified RNA oligonucleotides were prepared by in vitro transcription using T7 RNA polymerase from the corresponding DNA templates as described previously[13]. In short, a typical transcription reaction contained 1 μM DNA template, 4 mM of each NTP, 2 mM spermidine, 30 mM $MgCl_2$ and 10 mM DTT. To generate a 5′ monophosphate for subsequent ligation reactions, the reaction mixture was additionally supplemented with 15 mM GMP. After 4–6 h, the transcriptions were quenched by the addition of gel loading dye and purified on denaturing PAGE followed by extraction and precipitation with Ethanol.

## Optimization of random ligation

The reaction conditions for random ligation were optimized to increase the fraction of linear products and decrease the circular RNA formation. Starting from the manufacturers protocol (https://international.neb.com/protocols/2018/10/17/protocol-ligation-of-an-oligo-to-the-3-end-of-rna-using-t4-rna-ligase-1m0204), a) the reaction temperature was decreased from 25 °C to 4 °C, b) the reaction time was increased from 2 h to 6 h + 16 h, c) the ATP concentration was decreased from 1 mM to 0.1 mM, d) no benefit for higher PEG8000 concentration than 25% nor DMSO addition was identified and e) the oligo amount was increased from 40 pmol to 1000 pmol. The improvement in Random Ligation is shown in Supplementary Fig. 1.

## RNase R treatment

The relative proportion of circular product was determined by RNase R digestion, which degrades all linear RNA. 10 μl ligation reaction was digested with 1 μl RNase R (Lucigen, RNR07250) in a total volume of 20 μl for 30 min at 37 °C. The enzyme was inactivated 20 min at 65 °C. Comparable amounts of undigested and RNase R treated samples were analyzed on 10% TBE-urea gels (Thermo Fisher Scientific, EC6875BOX) (Supplementary Fig. 1b, d).

## Random ligation (RL) and polyadenylation

First, random ligation was optimized (see Supplementary Fig. 1). 1 nmol RNA oligos with 25% PEG8000 and 0.1 mM ATP in 1x T4 RNA ligase buffer and 10 U T4 RNA ligase 1 (New England Biolabs, M0437M) was ligated for 6 h on ice. After this, additional 10 U T4 RNA ligase 1 were added and ATP increased to 0.2 mM. Ligation was carried on overnight at 4 °C. Ligation products were purified with RNA Clean & Concentrator-5 (Zymo Research, R1016) according to the standard protocol. Then, ligation products that should be sequenced together were mixed at approximately equimolar ratio and polyadenylated with E-PAP Poly(A) Tailing Kit (Thermo Fisher Scientific, AM1350) in a total volume of 50 μl with 1.5 μl RiboLock (Thermo Fisher Scientific, EO0381) and 2.5 μl E-PAP for 5 min at 37 °C. Subsequently, polyadenylated mixtures were purified with RNA Clean & Concentrator kit-5 according to the standard protocol. Samples were either subjected directly to library preparation or stored at −80 °C until further usage.

## PAGE analysis of random ligation products

Aliquots of the ligation reaction were analyzed on 10% TBE-urea polyacrylamid gels (Thermo Fisher Scientific, EC6875BOX) to assess ligation efficiency. 2 μl ligation reaction or low range ssRNA ladder (New England Biolabs, N0364S) were heated for 5 min to 70 °C in 10 μl Novex TBE-urea sample buffer (Thermo Fisher Scientific, LC6876) and rapidly cooled before loading. Additionally, 1 μl GeneRuler Ultra Low Range DNA Ladder (Thermo Fisher Scientific, SM1213) was used for size estimation. Gels were stained 15 min at room temperature with SYBR Green II RNA gel stain (Thermo Fisher Scientific, S7564) and images acquired on a ChemiDoc Imaging System (Bio-Rad).

## Splinted ligation (SL)

For concatenation by splinted ligation, 500 pmol RNA (containing equal amounts of the individual RNA sequences) and 500 pmol of the DNA splint (ordered from Microsynth, see Supplementary Table 2) were annealed (5 min at 95 °C, 10 min at 25 °C) in annealing buffer (4 mM Tris-HCl (pH 8.0), 15 mM NaCl, 0.1 mM EDTA). Then, $MgCl_2$ was added to a final concentration of 8 mM along with 1.5 μl 10x T4 RNA ligase 2 buffer and 15 U of the T4 RNA ligase 2 (New England Biolabs, M0239S). Ligation reactions were carried out in a final reaction volume of 15 μl. The ligation reactions were incubated overnight at 25 °C, followed by loading on 15% denaturing PAGE (distributed into 5 wells on a 85 × 70 × 1 mm gel, 200 V, 50 min) with running buffer 1x TBE (89 mM Tris, 89 mM boric acid, 2 mM EDTA, pH 8.3). Synthetic RNAs of 35, 104 and 193 nts were used as size markers. The gels were stained with SYBR green I (Merck, S9430) and imaged on a ChemiDoc Imager (Bio-Rad). Ligation products > 150 nt in length were excised and extracted with TEN buffer (10 mM Tris-HCl, pH 8.0, 1 mM EDTA, 300 mM NaCl). The RNA was recovered by precipitation with cold ethanol, yielding around 20-25 ng ligated RNA product per 15 μl ligation reaction. Polyadenylation was done as for randomly ligated products described above.

## HEK293 RNA isolation and polyA⁺ RNA purification

HEK293 cells were obtained from the DSMZ – German Collection of Microorganisms and Cell Cultures GmbH (ACC 305) and cultured in DMEM (Thermo Fisher Scientific, 10566016) with 10% fetal bovine serum (Merck, F7524) and 1 x Penicillin/Streptomycin (Thermo Fisher Scientific, 15140122) at 37 °C, 5% $CO_2$. RNA was isolated with Trizol (Thermo Fisher Scientific, 15596026) according to the manufacturers protocol. PolyA⁺ RNA isolated as follows: For isolation of polyA⁺ RNA from HEK293 total RNA, 30 μg total RNA was digested with 1 μl DNase I (New England Biolabs, M0303S) in a total volume of 100 μl for 10 min at 37 °C followed by inactivation of the enzyme (10 min, 65 °C). 50 μl Dynabeads Oligo(dT)₂₅ beads (Thermo Fisher Scientific, 61002) were washed once with binding buffer (10 mM Tris pH 7.5, 1 M lithium chloride, 6.5 mM EDTA) and resuspended in 110 μl binding buffer. Beads and DNase I-treated RNA were combined and incubated 5 min at room temperature. Beads were collected on a magnet and washed two times with 200 μl washing buffer (5 mM Tris pH 7.5, 150 mM lithium chloride, 1 mM EDTA). RNA was eluted in 100 μl water by heating to 70 °C for 2 min. Beads were resuspended in 100 μl binding buffer and

recombined with the eluted RNA for a second purification round as described above. Final polyA$^+$ RNA was eluted in a volume of 10 μl.

## cDNA IVT synthesis

100 ng polyA$^+$ RNA was assembled in a total volume of 6 μl with 1 μl 10 μM RT primer (5′-TTTTTTTTTTTTTTTTTTTTTTTTTTTTTTVN-3′, IDT) and 1 μl 10 mM dNTPs (Thermo Fisher Scientific, R0191). The primer was annealed by incubation for 5 min at 75 °C, followed by 2 min at 42 °C. In a new tube, the template switching RT mix was prepared, composed of 2.5 μl 4x template switching RT buffer, 1 μl 10x template switching RT enzyme mix (New England Biolabs, M0446S) and 0.5 μl 20 μM template switching oligo (5′-ACTCTAATACGACTCACTA-TAGGGAGAGGGCrGrG+G-3′, IDT). RNA and RT mix were combined and incubated in a thermocycler (60 min, 42 °C; 10 min, 68 °C). For Second Strand Synthesis, 50 μl Q5 Hot Start High Fidelity Master Mix (New England Biolabs, M0494S), 5 μl RNase H (New England Biolabs, M0297), 2 μl 10 μM T7 extension primer (5′-GCTCTAATACGACTCACTATAGG-3′, IDT) and 33 μl H$_2$O were added. Second Strand Synthesis was carried out in a thermocycler (15 min, 37 °C; 1 min, 95 °C; 10 min, 65 °C). The product was cleaned up employing the NucleoSpin kit (Macherey Nagel, REF 740609.50) and subjected to in vitro transcription with the T7 MegaScript kit (Thermo Fisher Scientific, AM1333) according to the manufacturer's instructions. The generated IVTs were cleaned up on RNA Clean and Concentrator-25 (Zymo Research, R1017).

## Composition of ligation reactions and sequencing libraries

Random ligation oligos were concatenated to homopolymers for mAFiA training/validation. For every DRACH motif, a representative oligo was used either in the unmodified or m$^6$A-modified form. Oligo homopolymer pools composed of either unmodified (UNM) or m$^6$A-modified (MOD) oligos were generated from 2-4 sequences. These oligo pools were polyadenylated and subjected to direct RNA-sequencing. The respective sequencing runs are summarized in Supplementary Table 3. For TEST1 (Fig. 2b, c), the respective unmodified or modified oligos RL_M4_S0 and RL_M5_S0 were mixed in equimolar amounts and randomly ligated. All four possible combinations of unmodified and modified sequences were generated and sequenced individually (see Supplementary Table 3). The splinted ligations were performed separately for unmodified and m$^6$A-modified oligos in batches of three oligos. These were sequenced individually (2 sequencing runs of unmodified oligos and 2 sequencing runs of modified oligos, see Supplementary Table 3). For TEST2, the two oligos SL_AB and SL_BA were ligated in separate ligation reactions and sequenced as mix (Supplementary Table 3).

## Direct RNA-seq library preparation and ONT sequencing

Direct RNA-sequencing libraries were prepared from approximately 1 μg of polyadenylated oligo mixtures or 500 ng HEK293 WT/IVT polyA$^+$ RNA with RNA-SQK002 (Oxford Nanopore Technologies, ONT hereafter) with some modifications as previously described[14]. Ligation reactions were done for 10 min at room temperature. Libraries were quantified with the dsDNA HS Qubit kit (Thermo Fisher Scientific, Q32851) and loaded completely on Flongle or MinION flow cells (R9.4.1). Sequencing was performed for 24 h in MinION devices connected to a Dell workstation with high accuracy live basecalling.

## RODAN basecaller

The backbone basecaller is based on the fully-convolutional RODAN architecture[7] with several adaptations. First, the input length of the raw signal is shortened to 1024. Second, the decoder uses the Viterbi[15] rather than the default CTC algorithm[16]. Third, the model is trained on IVT HEK293 mRNA instead of a mixture of WT data from different species. Data preparation follows the walkthrough of the ONT software Taiyaki[17]. Training procedure follows the instructions on RODAN's repository (https://github.com/biodlab/RODAN).

## mAFiA module

After basecalling raw fast5 files with RODAN and aligning the results to the reference with minimap2.22, mAFiA receives as input a bed file which annotates all target A nucleotides in the reference sequence matching one of the DRACH motifs {GGACU, GGACA, GAACT, AGACT, GGACC, TGACT}. For each target nucleotide on a single read, a feature vector **x**, of length 768 is extracted from the last convolution layer ('convlayers.conv21') of RODAN. **x** is normalized by its maximum absolute value. A linear logistic model[18] is then applied to it to predict a probability between 0 and 1:

$$P(m^6A) = \sigma(\mathbf{w_i} \cdot \mathbf{x} + b_i) \tag{1}$$

where $\sigma$ is the sigmoid activation function, $\mathbf{w_i}$ is a 768-dimensional weight vector corresponding to the $i$ th motif ($i \in \{1,...,6\}$), and $b_i$ is a scalar. $\mathbf{w_i}$ and $b_i$ are parameters to be optimized by training on the feature vectors obtained from synthetic samples.

## Training on synthetic molecules

dRNA-Seq data from both random ligation (RL) and splint ligation (SL) samples are basecalled with RODAN. Reference oligo sequences are mapped locally to each read and then chained together to infer the ligated sequence in each RNA strand. Afterwards, each central A nucleotide in an oligo is classified into one of the six target DRACH motifs. At the read location corresponding to each of these A nucleotides, a 768-dimensional feature vector is extracted from the last convolution layer ('convlayers.conv21') of RODAN. Features from unmodified (UNM) samples carry the label $y = 0$, while those from modified (MOD) carry the label $y = 1$. In case of sample imbalance between UNM and MOD, the more numerous class is randomly subsampled to match the size of the smaller class. For each motif, the collected feature samples are split into 0.75-0.25 train-validation sets. The number of features collected in the RL and SL datasets are listed in Supplementary Tables 4 and 5 respectively.

Training of the logistic regression parameters, $\mathbf{w_i}$ and $b_i$, for the ith motif ($i \in \{1,...,6\}$), proceeds by minimizing the binary cross-entropy, ie, log-loss:

$$L = -\sum_n \left[ y_n \ln(p_n) + (1 - y_n) \ln(1 - p_n) \right] \tag{2}$$

where $y_n = 0 \vee 1$ is the label for sample $n$, and $p_n = \sigma(\mathbf{w_i} \cdot \mathbf{x_n} + b_i)$, equivalent to $P(m^6A)$ in the main text, is the corresponding modification probability predicted from feature $\mathbf{x_n}$. Optimization is performed with the python module sklearn, using the default L-BFGS solver[19] and maximum iterations 1000. Convergence is reached when the gradient change is below the default value of 1e-4, and is typically reached within 10 minutes on a common laptop. Feature extraction from RODAN requires a GPU machine and the extraction time depends on the total number of reads in the training sample.

The 0.75-0.25 train-validation split is resampled by 4-fold cross-validation, reserving a different 25% portion each time for validation and using the rest for training. In all cases, the validation AUCs differ within a ± 0.01 margin (Supplementary Table 6).

## Testing on synthetic molecules

In both TEST1 and TEST2, basecalled reads are aligned to reference sequences containing all possible random combinations of constituent oligos using minimap2.22. In TEST1a and TEST1b, $P(m^6A)$ are predicted for the A positions aligned to motifs GGACC and UGACU at the centre of each aligned oligo.

In TEST2, $P(m^6A)$ are predicted for the A positions aligned to GGACU at positions 13, 26, 39, ... of each read. In case of discontinuous alignment to the reference, the section of the read with the longest

continuous mapping is chosen. The position with the highest $P(m^6A)$ on a read is defined as the reference point. Positions at $2n \times 13$ nucleotides, ie, $0,26,52,...nts$ from the reference point are defined as "even", while those at $(2n+1) \times 13$ nucleotides, ie, $13,39,65,...$nts are defined as "odd". The precision-recall curve is calculated with the inferred ground-truth that the even positions on a read contain $m^6A$ and the odd ones contain A.

### Definition of site-level stoichiometry S

Given N reads aligned to a specific "A" location of a reference genome / transcriptome that matches one of the target kmers, the site-level stoichiometry $S$ is given by

$$S = \frac{1}{N}\sum_{n=1}^{N}\Theta\left(P_n(m^6A) - 0.5\right) \qquad (3)$$

where $P_n(m^6A)$ is the $m^6A$ modification probability of the nth read at the aligned position. The step function $\Theta$ is equal to 1 if $P_n(m^6A) \geq 0.5$, and 0 otherwise.

### Comparison between HEK293 WT and METTL3-KO

Raw fast5 reads on HEK293 METTL3 knock-out (KO) replicate 1 are downloaded from the European Nucleotide Archive: https://www.ebi.ac.uk/ena/browser/view/PRJEB40872. The data is basecalled with RODAN and aligned with minimap 2.22 to the human reference genome GRCh38. mAFiA site-level stoichiometric predictions are calculated on sites with minimum coverage 50. For the whole-chromosome $m^6A$ profile in Supplementary Fig. 4h, chromosome positions are binned into 10000 intervals, and the median S among all predicted sites in each interval is chosen to represent its characteristic methylation level.

### GLORI benchmark

dRNA-Seq reads are basecalled and then aligned to genome GRCh38. Reads with mapq below 50 are filtered out. Read-level $P_n(m^6A)$ is thresholded at 0.5 to classify each nucleotide on a read into modified or unmodified status. The site-level stoichiometry $S$ is then calculated by taking the fraction of modified nucleotides aligned to a specific site (see also definition of $S$ above). A minimum dRNA-Seq coverage of 50 is required for a site to be compared against GLORI measurements. A more detailed analysis of mAFiA's correlation with GLORI at various coverage and stoichiometry is shown in Supplementary Fig. S4a–c.

### Testing on *Arabidopsis*

Raw fast5 data obtained from *Arabidopsis thaliana*, strain *col0* replicate 1 and strain *vir1* replicate 1, are downloaded from the public database https://www.ebi.ac.uk/ena/browser/view/PRJEB32782. Reads are basecalled with RODAN and aligned to the TAIR10 genome reference (https://www.arabidopsis.org/) with minimap2.22. miCLIP2 sites are obtained from supplementary data of Parker et al.[9].

### Computational requirement and run-time

Testing on HEK293 and *Arabidopsis* samples are performed on computing cluster nodes, each with 2 or 4 cpu cores, 80GB memory, and an Nvidia Quadro RTX 6000 GPU.

Software and models are available at: https://github.com/dieterich-lab/mAFiA. For the example given in the walkthrough (GLORI sites on chromosome X of HEK293 WT), basecalling with RODAN takes about 30 minutes, and mAFiA (including feature extraction from RODAN) takes about 45 minutes.

### Comparison to CHEUI and m6Anet

Basecalling for CHEUI and m6Anet is done with ONT Guppy 6.4.6. The results are mapped to the reference with minimap 2.22. Resquiggling is done with nanopolish 0.13.3. The remaining procedures follow the documentations of CHEUI and m6Anet 2.0.1. Read-level predictions are used for TEST1 and TEST2, while site-level predictions are used for HEK293. For GLORI sites that map to multiple transcripts, a coverage-weighted average of the transcript-specific predictions is performed to produce a single stoichiometry for each genomic coordinate. Transcriptome coordinates from GRCh38.102 are converted into genomic coordinates with the Ensembl Rest API (https://rest.ensembl.org/). Additional comparison of the methods using miCLiP as ground-truth is shown in Supplementary Fig. S4d.

### Reporting summary

Further information on research design is available in the Nature Portfolio Reporting Summary linked to this article.

## Data availability

The oligo training and validation data as well as HEK293 WT dRNA data generated in this study have been deposited in the European Nucleotide Archive under accession number PRJEB74106. Because of the large size of the data sets, HEK293 WT/IVT mixing data will be made available upon request. Other publicly available data used in this manuscript: HEK293 METTL3-KO: https://www.ebi.ac.uk/ena/browser/view/PRJEB40872. *Arabidopsis*: https://www.ebi.ac.uk/ena/browser/view/PRJEB32782. Source data are provided with this paper.

## Code availability

Software[20] and models are available at: https://github.com/dieterich-lab/mAFiA.

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

## Acknowledgements

This work was supported by funding from the Klaus Tschira Foundation (grant no. 00.013.2021), the Heidelberg AI Health Cluster (grant no. 32-7533-6-1218.3/3/12/4) and the German Science foundation (DFG TRR319 RMaP grant no. 439669440) to CD. We are grateful to Fred Hamprecht for providing computing time at the Interdisciplinary Center for Scientific Computing in Heidelberg.

## Author contributions

AC developed the computational methods. CS synthesized and ligated splint RNA oligos. IN performed all random ligations, HEK293 experiments and ONT sequencing experiments. CH and CD supervised the project. CD conceptualized the project. AC, IN and CD wrote the manuscript. All authors revised the manuscript.

## Funding

## Competing interests

The authors declare no competing interests.
