## [Peer Review File · Nature Communications]

Detecting m6A at single-molecular resolution via direct-RNA sequencing and realistic training dataREVIEWER COMMENTS

Reviewer #1 (Remarks to the Author):

Chan et al. report on the synthesis of training data with nucleotide modifications and an algorithm for the detection of synthetic and natural m6A modifications.

The data set and the algorithm are very interesting, timely and needed. However, the evaluation is very difficult, due to the lack of explanations. Therefore, I ask the reviewers kindly to improve the manuscript by the explanation of the following terms and ideas.

- 1) Please introduce your abbreviations carefully (e.g. RL, SPL, ...)
- 2) a table presenting absolute numbers would be nice (see histogram; how many of how many sites have been called correctly? possibly give a confusion matrix)
- 3) I do not understand Figure 1g. Please improve the figure and explain it.
- 4) Why Figure 1j has been used remains unclear to me
- 5) I do not understand Figure 1f. Please improve the figure and explain it. The plot shows $P(m6A)$ for $0.8 \leq P \leq 1.0$; if the threshold for m6A is at 0.5, then all positions have been called in the plot? The limit for the heatmap should be adjusted (range from 0.0 to 1.0)
- 6) similar to Figure 1h (heatmap)
- 7) Please also explain the four fold symmetry
- 8) An explanation why the introduced 'K' as comparison for the methods is used. Please provide additionally usual measures, such as accuracy, specificity, f1, recall, MSE
- 9) Please specify details to the transfer learning method.
For evaluation the effectiveness of transfer learning can be given by comparing the performance of the transfer learning (pre-trained) vs. a baseline model (trained from scratch)
Possibly the transfer learning is not necessary?
Please provide the convergence speed.
- 10) Please provide standard details, such as validation percentage (next to 75% trainings and 25% test data); how often it has been cross evaluated?
Please check also for robustness (e.g. by adversarial attack or noise injection)
- 11) What are the training time, inference time and memory usage?
That might be crucial for a broader application.

Reviewer #2 (Remarks to the Author):

m6A, considered the most prevalent epigenetic modification in RNA, plays a significant role in regulating various functions of cellular RNA species. However, accurate identification of m6A and assessment of its stoichiometry across the entire transcriptome have presented significant challenges. Traditionally, exploration of the m6A landscape primarily relied on experimental methods based on next-generation sequencing platforms. In recent years, the advent of direct RNA sequencing technology based on the Oxford Nanopore Technologies (ONT) platform has spurred the development of diverse computational tools, providing new alternatives for m6A detection and quantification.

However, the lack of suitable training data with modifications severely limits the development of ONT tools. In this manuscript, the authors attempt to address the issue by creating realistic training data. Short oligoRNAs with specific m6A motifs were synthesized and concatenated using various ligation methods to construct two distinct training data sets. Subsequently, they developed an algorithm, mAFiA, to identify m6A at single-molecular resolution from nanopore data. Comparisons between mAFiA and a recently reported experimental method for absolute quantitative detection of m6A (GLORI) demonstrate mAFiA's superior performance over other similar computational tools. To enhance the manuscript, it is recommended to provide more detailed prediction and analysis results for whole-transcript m6A, showcasing mAFiA's detection capabilities across various conditions. Additionally, a comprehensive comparison of mAFiA with existing technologies (both experimental methods and computational tools) would further emphasize its advantages and shed light on potential areas for improvement.

Major comments:

1. The accuracy of mAFiA in detecting various motifs differs significantly, notably in the case of the GAACU sequence. Please elucidate the reasons behind these variations. Additionally, is mAFiA capable of detecting other motif sequences aside from the six m6A motifs outlined in the article?
2. mAFiA appears to detect m6A and its stoichiometry. However, a critical consideration is its applicability to various biological samples. Incorporate the results of mAFiA's whole transcriptome m6A analysis for a specific cell line, like WT HEK293T cells. Additionally, please validate the accuracy of mAFiA through more direct methods, such as comparing m6A profiles in the whole transcriptome between WT and METTL3 KO samples.
3. Figure 2d suggests that mAFiA exhibits a less-than-optimal sensitivity for identifying low-abundance m6A sites. I'm interested in understanding the count of m6A sites that mAFiA can reliably detect in the WT HEK293T transcriptome. Additionally, please outline the range of m6A methylation levels that could be predicted by mAFiA. The ability of mAFiA to quantify transcriptome-wide m6A across various motifs should also be determined.
4. The performance of computational tools for direct RNA sequencing can be influenced by factors like sequence coverage and m6A stoichiometry. It's essential to check if mAFiA is affected similarly.
5. A thorough comparison of mAFiA's results with that of existing experimental methods (e.g., GLORI, miCLIP or MAZTER-seq) and other ONT tools (e.g., xPore, Nanom6A, m6Anet or EpiNano) is necessary.

Minor comments:

1. Please clarify the exact meaning of "non-intersecting" mentioned on page 1, line 30.
2. What is the specific definition of P(m6A) in the manuscript? Can it represent the predicted level of m6A? Describe it more clearly in the text.
3. Six m6A motif sequences were selected for the training data set (RL, SPL). Are the modification levels in each m6A motif determined and specified? Additionally, details about the composition of the training datasets are not provided. Please add it to the main text or supplementary information of the manuscript.

4. In the legend of Figure S1b, does 'RL-M1S0' correspond to the sequence 'M1_S0' in Table 1? Standardizing the names would enhance clarity.

5. Figure S2 displays a gel map example. However, significantly different lengths of oligoRNAs using two ligation methods are observed. Could these differences potentially impact the training and subsequent prediction results of mAFIA? Additionally, I noticed the variations in length and concentration of the final ligation products from similar sequences (e.g., M0_S0 and M1_S0). Please explain these discrepancies.

6. The training data was constructed using M3_S0, M4_S0, and M5_S0 by splint ligation. Does this data set only have three different m6A motifs? Are there any special considerations for this design? If not, please supplement relevant details with the manuscript.

7. Please describe the results for each figure within the figure legends in detail.

Reviewer #3 (Remarks to the Author):

In the manuscript, Chan et al describe a new method to detect m6A from nanopore direct RNA-Seq data for individual RNA reads. To be able to train the model, the authors generated new data such that the modification status of each read is known. The authors used synthetic sequences that contain a single m6A site (or non m6A site) surrounded by 21-33 bp unmodified RNA sequence from the human transcriptome corresponding to known m6A positions. The data contains 6 different DRACH motifs, capturing 80% of human DRACH sites, and the different sequences are combined using two strategies. The computational method that is then introduced uses these data to train a logistic regression classifier that uses the features from the RODAN basecaller to predict m6A for individual reads. The method is evaluated using synthetic sequences and human RNA, and compared against other methods that are able to perform single molecule m6a prediction.

The results show that the proposed method has a high accuracy and a high similarity with GLORI-Seq results, suggesting that the proposed approach to train a single molecule m6a classifier using the newly generated training data provides an advantage.

The training data that is described in this study is different from what others have used and address an important point by enabling the training of a single molecule classifier. The manuscript is short, and overall well written and sufficiently clear. I have some concerns as outlined below:

Major comments:

(1) The data, the training procedure, and the evaluation procedure are not sufficiently described

From the manuscript it was not sufficiently clear how the training data is generated, how many samples were generated, and how the data was used for model training and evaluation:

- The data is currently not available (or at least I could not find any link in the manuscript). Please make the data available

- It is not clear how modified and unmodified data are generated. Did the authors generate different libraries using the unmodified and modified oligos? How many replicates were generated? From the current description it is unclear how the authors obtain data with labels for each read as the description is currently missing.

- is one model learned for all kmers, or do the authors train a separate model for each kmer?

- how does the model perform when reads from different kmers are analysed together (i.e. one precision recall curve for all kmers)? Such an analysis could help to test if some kmers will be ranked lower than others, which would impact results for example on human data where all kmers are

analysed together

- The method is evaluated on a balanced data set it seems. How are the two classes defined? Is the data filtered in any way to obtain balanced data?
- the oligo sequences contain several positions with A kmers that are not at the center, and which could be used as negative data points (for training or just for testing). Does the model accurately discriminate the unmodified (non center) kmer/positions at reads from the modified (center) kmer/positions at reads?

(2) Comparison with other methods for single molecule accuracy

- The authors evaluate their own model using Precision Recall curves. However, the comparison with other methods is only done using the AB|BA design. The authors should also compare other methods using Precision-Recall curves on the same data set.
- I did not fully understand the AB|BA AB-BA design, or why this was used to evaluate single molecule predictions. I assume this is related to the design of the sequences which might be either including all modified center positions, or all unmodified center positions, so the authors designed this experiment to have reads with both modified and unmodified positions while still having approximate labels. This experiment could be better motivated and explained.
- It's unclear why the AB-BA/AB|BA experiment is used to compare mAFIA with other methods, as the evaluation depends on selecting a threshold for the read probability, whereas the precision recall would not require to select a threshold (see comment above)
- The authors choose a threshold based on the range of 0.1 to 0.9. This range will cover most probability values for a classifier learned on balanced data, where the expected probability might be 0.5 (as in Figure 1c). However, it's not clear if that range covers CHEUI and m6anet as well. For completeness, the authors should evaluate values between 0 and 0.1 and 0.9 and 1 as well.
- Another control is to compare the modification ratios that are calculated from m6anet and CHEUI across all reads and the m6a probability for the position (not just the single molecule predictions) with mAFIA for the AB-BA/AB|BA experiment.

(3) prediction of m6A sites (not single molecule predictions)

- While the ability to predict single molecules with mAFIA using the training data design is clearly a novelty, ultimately the ability to detect which positions are modified will still be important for most use cases. Currently this aspect is not evaluated. How well can the authors identify m6A sites? This evaluation could be done with the synthetic sequences by combining all data from all motifs (which will be a balanced data set), and in comparison with the GLORI-Seq results where many more unmodified sites exist and where the kmers have different frequencies, resulting in an unbalanced but more realistic scenario. How is the precision-recall in these 2 scenarios?
- Currently it is not clear if the model can handle unbalanced data with a much larger number of unmodified positions. These positions seem to be excluded from the comparison with GLORI-Seq (p -value < 0.001). How does mAFIA stoichiometry results look for positions where mAFIA is confident (>50 reads) but GLORI-Seq is not confident?
- CHEUI and m6anet make site level predictions, how do the site level predictions compare against the predictions from mAFIA for individual positions? This comparison can again be done using the precision-recall to avoid having to select a threshold

(4) Identification of m6a in human and non-human data/sequence context

- The model is trained on data that captures the complexity of human m6A sites. However, other species such as yeast or arabidopsis, or synthetic data will have a different m6A motifs, and it is not clear how well the model will work on such data. How well does their method identify m6A in non-

human sequences? If the model does not identify m6A in non-human RNA, or if it only captures a small percentage of sites due to the specific motif selection, then this should be clearly described to avoid users applying it to data that is not supported by this model.

Minor:

-figure 1c: How was the data normalised? How do these figures look without normalisation?

- Figures 1f and 1h are unclear. What is the x-axis? Are all positions predicted to have $P(\text{m6A}) = 0.8$?

- Figure 1a, the schematic does not sufficiently explain the design. What are the colours? WHICH sequences have m6a, which don't have m6a? A clearer illustration would be very helpful (see also point 1)

- "Unlike other single-molecule methods, mAFIA does not require additional post-processing such as nanopolish7, and can be integrated into an existing basecaller without altering the latter's accuracy." I assume that most nanopore users by default get the basecalled sequences with the nanopore basecaller. The mAFIA method requires users to run a different basecaller (RODAN), which also requires resources in addition to the nanopore basecalling then. If the authors want to keep the claim that their method has minimal computational overhead, then they should compare the resource requirements for RODAN vs Nanopolish, or the authors use features from the default output of nanopore basecalling. Alternatively, I would suggest to specify these claims to make clear that this is only true when the RODAN basecaller is used.

- Related to this, since the RODAN basecaller seems to be essential to the manuscript, I would recommend to mention RODAN in figure 1b, in the main text, and to cite the paper in the main text references to fully acknowledge the method.

Links for reviewers

Oligo:

https://zenodo.org/records/8318848?token=eyJhbGciOiJIUzUxMiJ9.eyJpZCI6Ijg2ZjQzMDA0LWQyYzUtNDZhNS1hZWZjYjYxNWY4NjVlZiIsImRhdGEiOnt9LCJyYW5kb20iOiJzMTY5MDk2NjIzYmUzMTQ0OQWYwZDkyNjAyYTI3ZDU0OCJ9.PVucSvl91DU9jGnBqqOHBXbgjivo-5IPkjniHFSIcvuYgzVbVVR7if5zL8e6hq6e_DIZ27WJC39qIYeJ3VHnLtA

HEK293 WT:

https://zenodo.org/records/8319583?token=eyJhbGciOiJIUzUxMiJ9.eyJpZCI6IjA0OGFiMGU5LTVhMzItNDcxOC1iYzQxLWZjYjYxNWY4NjVlZiIsImRhdGEiOnt9LCJyYW5kb20iOiJmMTI3Y2MONTgxYjc3MjJmZTNmMmI0N2M0ZTczZDMwZSJ9.Dd_VrrtrcE1yoctln2AKy_DTgRXsTDbD5-yp3zZqlD4L0OyAjPu1J2JnjWljwFDutFIHgf7LcjGmJfcdjGcVyw

Reviewer #1 (Remarks to the Author):

Chan et al. report on the synthesis of training data with nucleotide modifications and an algorithm for the detection of synthetic and natural m6A modifications.

The data set and the algorithm are very interesting, timely and needed. However, the evaluation is very difficult, due to the lack of explanations. Therefore, I ask the reviewers kindly to improve the manuscript by the explanation of the following terms and ideas.

1) Please introduce your abbreviations carefully (e.g. RL, SPL, ...)

Response: We have now introduced the abbreviations for random ligation (RL) and splint ligation (SL) in the second paragraph of the main text, and sketched the difference between the two approaches in Figure 1a and its accompanying caption. The chemical details that differentiate the two synthetic approaches are described in the Supplementary Methods section.

2) a table presenting absolute numbers would be nice (see histogram; how many of how many sites have been called correctly? Possibly give a confusion matrix)

Response: Absolute numbers of training and validation samples are now tabulated in Supplementary Tables T4 (RL) and T5 (SL). A confusion matrix corresponding to the results of Figures 1c, d is now given in Figure S3a.

3) I do not understand Figure 1g. Please improve the figure and explain it.

Response: We apologize for the unclear representation of the results in the previous Figures 1f,g and removed these. The design and results of this experiment are now presented in the new Figures 1e, 1f, 1g. Figure 1e shows the experimental design of sequencing random polymers composed of one methylated and one unmethylated DRACH motif oligo (TEST1). The results are presented in Figure 1f as violin plots along with a detailed caption. Furthermore, we compare our results in Figure 1g to CHEUI and m6Anet predictions.

4) Why Figure 1j has been used remains unclear to me

Response: The previous Figure 1j was intended to show the contrast between regularly spaced modification patterns versus a randomized baseline. We have now simplified the explanation and added an illustration of the RNA molecules in Figure 1e (bottom, TEST2). The detected m⁶A patterns are visualized with IGV in Figure 1h. The results of this test scenario are shown as violin plot in Figure 1i and comparison to other methods in Figure 1j.

5) I do not understand Figure 1f. Please improve the figure and explain it. The plot shows $P(m^6A)$ for $0.8 \leq P \leq 1.0$; if the threshold for m⁶A is at 0.5, then all positions have been called in the plot? The limit for the heatmap should be adjusted (range from 0.0 to 1.0)

Response: Only the A positions in GGACC and UGACU are called in the plot. We only intended to show the spacing between the two 5mers on a read-by-read basis. To avoid confusion, this plot is now removed, and we only show the overall distribution of $P(m^6A)$ at the relevant positions as violin plots in the new Figure 1f.

6) similar to Figure 1h (heatmap)

Response: This is now replaced with a new Figure 1h, in an IGV snapshot, which shows the variation of the predicted $P(m^6A)$ at regularly spaced positions, on a per-read basis. Readers can now clearly see that the nucleotides in between the relevant positions are either not A nucleotides, or A nucleotides that do not match our target DRACH 5mers. Furthermore, we now include a violin plot of the $P(m^6A)$ in the new Figure 1i.

7) Please also explain the four fold symmetry

We realize that this terminology can be confusing, and replaced it with a more detailed explanation in the main text of how the predicted $P(m^6A)$ in two different 5mers show mirroring distributions in TEST1a and TEST1b. In the caption of Figure 1f, we added this sentence: “The $P(m^6A)$ contrast between the two motifs is flipped in TEST1b, ...”

8) An explanation why the introduced 'K' as comparison for the methods is used. Please provide additionally usual measures, such as accuracy, specificity, f1, recall, MSE

Response: The contrast parameter K was introduced to compare a regularly spaced pattern against a random baseline. In the revised manuscript, we have simplified the presentation of our experiment and switched to a more commonly known metric instead, namely area-under-precision-recall-curve (AUPRC). The same metric is now used in all our benchmarks (internal validation, TEST1 and TEST2) to compare against other methods, without the need to select a specific threshold for $P(m^6A)$.

9) Please specify details to the transfer learning method. For evaluation the effectiveness of transfer learning can be given by comparing the performance of the transfer learning (pre-trained) vs. a baseline model (trained from scratch) Possibly the transfer learning is not necessary? Please provide the convergence speed.

Response: We realize that the term “transfer learning” could mislead readers into thinking that we retrained the original basecaller architecture with new classes. In fact, our method simply extracts features from the last convolutional layer of the backbone network and feeds them into an entirely different classifier. In this sense, the basecaller itself is never altered. It always predicts four classes (A, C, G, U).

To make the idea explicit, we have added this to the 3rd paragraph of the main text: “Our approach does not interfere with the accuracy of the original basecaller, ... without the need to retrain the entire set of models from scratch.”

We have also added more details about the method in the Supplement (“mAFiA module”). The convergence speed is covered in the Methods section “Training on synthetic molecules”.

10) Please provide standard details, such as validation percentage (next to 75%trainings and 25% test data); how often it has been cross evaluated? Please check also for robustness (e.g. by adversarial attack or noise injection)

Response: The 25% reserved in each dataset is indeed the validation data. To avoid ambiguity, we now call the 75-25 split train-validation instead. The real test data that we use to compare against other methods is of entirely different composition, and is totally unseen during training and validation. We now call them TEST1 and TEST2.

We have evaluated our sampling through 4-fold cross-validation, i.e., reserving a different 25% portion for validation each time and using the rest for training. In all cases, the average AUPRC differs by less than 0.01 (Table T6). We hence conclude that the training procedure is robust to sampling fluctuations. This is added to the methods section “Training on synthetic molecules”.

Regarding noise injection, we believe that the best demonstration of model robustness against *naturally occurring* noise is to test it against multiple datasets sourced from various conditions. Our two oligo datasets, RL and SL, are generated in two different laboratories, using different enzymatic synthesis. As for natural mRNA (HEK293 WT / IVT, METTL3-KO, Arabidopsis), we have tested our method on close to 10 million reads in total, sequenced in at least 3 different laboratories around the world, involving both human and plant species. We are confident that the span and variety of these samples indeed encapsulate the noise that the sequencing technology can possibly encounter.

11) What are the training time, inference time and memory usage? That might be crucial for a broader application.

Response: We have added a supplementary section “Computational requirement and run-time” that discusses this topic:

“Testing on HEK293 and *Arabidopsis* samples are performed on computing cluster nodes, each with 2 or 4 cpu cores, 80GB memory, and an Nvidia Quadro RTX 6000 GPU.

Software and models are available at: <https://github.com/dieterich-lab/mAFiA>. For the example given in the walkthrough (GLORI sites on chromosome X of HEK293 WT), basecalling with RODAN takes about 30 minutes, and mAFiA (including feature extraction from RODAN) takes about 45 minutes.”

Reviewer #2 (Remarks to the Author):

m6A, considered the most prevalent epigenetic modification in RNA, plays a significant role in regulating various functions of cellular RNA species. However, accurate identification of m6A and assessment of its stoichiometry across the entire transcriptome have presented significant challenges. Traditionally, exploration of the m6A landscape primarily relied on experimental

methods based on next-generation sequencing platforms. In recent years, the advent of direct RNA sequencing technology based on the Oxford Nanopore Technologies (ONT) platform has spurred the development of diverse computational tools, providing new alternatives for m⁶A detection and quantification.

However, the lack of suitable training data with modifications severely limits the development of ONT tools. In this manuscript, the authors attempt to address the issue by creating realistic training data. Short oligoRNAs with specific m⁶A motifs were synthesized and concatenated using various ligation methods to construct two distinct training data sets. Subsequently, they developed an algorithm, mAFiA, to identify m⁶A at single-molecular resolution from nanopore data. Comparisons between mAFiA and a recently reported experimental method for absolute quantitative detection of m⁶A (GLORI) demonstrate mAFiA's superior performance over other similar computational tools. To enhance the manuscript, it is recommended to provide more detailed prediction and analysis results for whole-transcript m⁶A, showcasing mAFiA's detection capabilities across various conditions. Additionally, a comprehensive comparison of mAFiA with existing technologies (both experimental methods and computational tools) would further emphasize its advantages and shed light on potential areas for improvement.

Major comments:

1. The accuracy of mAFiA in detecting various motifs differs significantly, notably in the case of the GAACU sequence. Please elucidate the reasons behind these variations. Additionally, is mAFiA capable of detecting other motif sequences aside from the six m⁶A motifs outlined in the article?

Response: We would like to emphasize that mAFiA's accuracy on five out of the six DRACH motifs is high and comparable (AUC ≥ 0.94). However, we noted as well that its precision-recall for the GAACU motif is lower (AUC = 0.86). We hypothesize that this is caused by a higher error rate in alignment due to the two consecutive As, which may cause a mis-location of the central nucleotide. This is likely to be a generic issue affecting all ONT methods, as we observe that CHEUI and m6Anet also perform worse on GAACU than on other motifs (please refer to Figures S4e and S4f). Comparing all 3 methods on GAACU only, mAFiA still has a much higher correlation with GLORI (0.80, Figure 2d) than CHEUI (0.35) and m6Anet (0.59).

As described in the updated methods sections "mAFiA module" and "Training on synthetic molecules", each classifier is optimized to detect m⁶A pattern in a specific 5mer. The training procedure for any additional motif is independent of others, and we continuously expand our set of models as new training data becomes available.

We would like to reiterate that the primary goal of this brief communication letter is to demonstrate the technical possibility of synthesizing and utilizing oligo RNAs that mimic naturalistic m⁶A sites in the human transcriptome. As such, we have chosen to demonstrate our method on the six most representative motifs, which cover a vast majority (~80%) of the consensus m⁶A sites in HEK293 cells. On these six motifs, we have demonstrated superior performance to other computational methods that are purportedly more exhaustive. Our technique can be easily extended to include the next 12 DRACH motifs (accounting for the remaining 20% of sites), but the additional-benefit-to-cost ratio was prohibitive for our budget.

2. mAFiA appears to detect m⁶A and its stoichiometry. However, a critical consideration is its applicability to various biological samples. Incorporate the results of mAFiA's whole transcriptome

m6A analysis for a specific cell line, like WT HEK293T cells . Additionally, please validate the accuracy of mAFiA through more direct methods, such as comparing m6A profiles in the whole transcriptome between WT and METTL3 KO samples.

Response: We have performed a whole-transcriptome analysis of HEK293, and compared the m⁶A profile in HEK293 WT versus METTL3 KO. In Figures 2c and S4h, we show that our method indeed captures transcriptome-wide down-regulation of m⁶A methylation in the KO sample. However, we would like to emphasize that the existence of full METTL3 KO are at least debated in the field and may not lead to a complete loss of m⁶A in mRNA. Thus, the IVT samples that are presented in Figures 2e-f provide cleaner datasets.

Additionally, we now include the analysis of an *Arabidopsis thaliana* data set (WT vs. *vir1* KO) in Figures 2g,h and show that mAFiA is capable of detecting m⁶A sites depleted in the *vir1* KO as well.

3. Figure 2d suggests that mAFiA exhibits a less-than-optimal sensitivity for identifying low-abundance m6A sites . I'm interested in understanding the count of m6A sites that mAFiA can reliably detect in the WT HEK293T transcriptome. Additionally, please outline the range of m6A methylation levels that could be predicted by mAFiA. The ability of mAFiA to quantify transcriptome-wide m6A across various motifs should also be determined.

Response: Please refer to Figure S4a for a break-down of mAFiA's site counts by stoichiometry in HEK293 WT. Our method can predict methylation levels ranging from 0 to 100%. Indeed, most of the site-level predictions it gives for the HEK293 cell line is in the low S range. The reason why there exists a gap below S=10% in the previous Figure 2d (now Figure 2e) is because GLORI does not publish site-level stoichiometries below 10%. We have now added this disclaimer explicitly in the main text: "...numerical values published by GLORI-Seq⁸, S_{GLORI} , which range from 10% to 100%."

The total count of m⁶A sites detected by mAFiA, and indeed any Nanopore-based method, depends on how many sites are sufficiently covered in the data. We have performed an analysis of mAFiA's overlap with miCLIP and m6Anet, but limiting to those sites where our HEK293 dataset indeed has sufficient coverage. Figure S4d shows that the sites predicted by mAFiA overlap to a large extent with those found by miCLIP. Compared at the same precision, mAFiA also recalls many more miCLIP sites compared to m6Anet and CHEUI.

On a transcriptome-wide level, we have also compared mAFiA predictions on close to 6000 sites on all chromosomes, with site-level methylation levels ranging from 10% to 100%, as published by GLORI, and found good correlations between the two sets of stoichiometries (Figure 2d). Furthermore, we have performed mixing experiments with HEK293 WT and IVT, and observed quantitative agreement between what mAFiA measures and the reduced methylation levels at each individual site as predicted. We believe this is clear evidence that our method works across a wide range of stoichiometries and across many locations on the transcriptome.

4. The performance of computational tools for direct RNA sequencing can be influenced by factors like sequence coverage and m6A stoichiometry. It's essential to check if mAFiA is affected similarly.

Response: We have added supplementary Figures S4b, c to study how mAFiA performance changes with coverage requirement and stoichiometry range. Using correlation to GLORI values

as a metric, we determine that a minimum coverage of 50-60 reads on a site is optimal for the accuracy of our stoichiometry predictions. In terms of the range of stoichiometry, the root-mean-squared difference with GLORI (lower RMS = higher correspondence) is largely stable above $S=30\%$, and shows a larger difference in the low range. However, we note that the authors of the GLORI manuscript have also stated that their method is not so accurate when S is low. Indeed, their published data does not include sites with $S<10\%$, and the sites with low S that are indeed published also have higher P -values (less confidence).

5. A thorough comparison of mAFiA's results with that of existing experimental methods (e.g., GLORI, miCLIP or MAZTER-seq) and other ONT tools (e.g., xPore, Nanom6A, m6Anet or EpiNano) is necessary.

Response: Experimentally, we have included comparison with GLORI and miCLIP in HEK293 (Figures 2d-f, S4d), as well as miCLIP in Arabidopsis (Figure 2g as well as numbers stated in main text). We have shown good agreement between mAFiA predictions and those orthogonal methods, both numerically (GLORI) and in terms of site overlap (miCLIP). Computationally, we have compared our results against m6Anet and CHEUI, which are the only ONT tools able to predict m6A on a single-molecule basis and to produce quantitative estimates of site-level stoichiometry. Indeed, the numerical benchmark we tabulated to compare various methods in Supplementary Figure S4g explicitly requires that a method can produce predictions on a read-by-read basis. To the best of our knowledge, xPore, Nanom6A, and EpiNano all predict the statistical distribution of site methylation on the aggregate level, and hence cannot be compared against the other 3 methods in the single-molecule benchmark.

Minor comments:

1. Please clarify the exact meaning of "non-intersecting" mentioned on page 1, line 30.

Response: This word is removed from the main text as it is not descriptive enough. Instead, we have tried to clarify the difference between the random ligation (RL) and splint ligation (SL) sequences by explicitly color-coding them in Table T1. The two sequences representing one DRACH motif share the same central 5mer, while the flank sequences differ.

2. What is the specific definition of $P(m^6A)$ in the manuscript? Can it represent the predicted level of m6A? Describe it more clearly in the text.

Response: $P(m^6A)$ is the modification probability of a single nucleotide on a single read, while the stoichiometry S of a site is the percentage of modified nucleotides aggregated from all the reads aligned to that site. Here, sites with $P(m^6A) \geq 0.5$ are considered as modified. To clarify the distinction between read-level and site-level predictions, we have now included a more detailed explanation in paragraph 7 of the main text. In the methods section, there is an explicit definition of the site-level stoichiometry S , and how it is calculated from the read-level $P(m^6A)$. Also, an IGV snapshot and histograms at different transcript locations (Figures 2a, b) illustrate the alignment of modified nucleotides and the aggregation of read-level $P(m^6A)$ into site-level S .

3. Six m6A motif sequences were selected for the training data set (RL, SPL). Are the modification levels in each m6A motif determined and specified? Additionally, details about the composition of the training datasets are not provided. Please add it to the main text or supplementary information of the manuscript.

Response: We would like to emphasize that we have used synthetic RNA oligos for training and validation. Thus, for every nucleotide the modification status is known. More specifically, all positions are unmodified, except for the central “A” in the modified RNA oligos, which is 100% m⁶A.

The 6 motifs chosen are the most prevalent in human mRNA according to previous studies with miCLIP. We have briefly mentioned this in the second paragraph of the main text: “...the six most common DRACH motifs, which collectively account for almost 80% of all consensus m⁶A sites in human mRNA⁴⁻⁶”. Analysis of GLORI-Seq leads to a similar conclusion. Since the natural modification levels of those motifs have already been thoroughly analyzed in the cited references, we refer the reader to those publications for more details.

We have listed all 15 training sequences in Table T1, along with the chromosomal locations from which they are sourced. We have also added more details to the ligation schematic in Figure 1a to explain the composition of the datasets and extended the Method section accordingly.

4. In the legend of Figure S1b, does ‘RL-M1S0’ correspond to the sequence ‘M1_S0’ in Table 1? Standardizing the names would enhance clarity.

Response: We apologize for these inconsistencies. We now have standardized the name to “RL_M0_S0” for oligos used in random ligation and “SL_M0_S0” for oligos used in splint ligation.

5. Figure S2 displays a gel map example. However, significantly different lengths of oligoRNAs using two ligation methods are observed. Could these differences potentially impact the training and subsequent prediction results of mAFiA? Additionally, I noticed the variations in length and concentration of the final ligation products from similar sequences (e.g., M0_S0 and M1_S0). Please explain these discrepancies.

Response: We noticed the differences in length of ligation products for the random ligation approach as well. While we have not analyzed the causes for the differences in ligation efficiency systematically, some sequences seem to be more prone to circularization (i.e. intramolecular ligation) than others. By addition of a polyA tail, we only select linear products for sequencing.

The length of the ligation products may indirectly affect the sequencing efficiency for a specific motif, as short reads may be classified as “failed”. As we aimed to only work with high quality data, we only used “passed” reads throughout and repeated the sequencing experiment where necessary to ensure that there are sufficient samples for training.

The mapping and subsequent training / predictions should not be influenced by the differences in length of ligation products, as we use a local alignment strategy (see Methods “Training on synthetic molecules” for details). This is further supported by the fact that, for example, the predictions for GGACU (M0_S0) and GGACA (M1_S0) have comparable accuracies although the ligation products are of different lengths.

We would like to note that RL_M0_S0 (UGCACAGAGGA/m6ACAAGUAGCUG) and RL_M1_S0 (UACUGAAGGAA/m6ACUGACCUCUC) are not similar sequences and differences in ligation efficiency may arise from different nucleotides at the 5’ and 3’ ends or secondary structures.

6. The training data was constructed using M3_S0, M4_S0, and M5_S0 by splint ligation. Does this data set only have three different m⁶A motifs? Are there any special considerations for this design? If not, please supplement relevant details with the manuscript.

Response: The splint ligation (SL) dataset contains oligos representing 6 DRACH motifs. In each library, 3 different oligos are splint-ligated into the same heteropolymer strand, resulting in two different libraries. Furthermore, each library is generated as an unmodified or modified variant. The random ligation (RL) dataset contains the same 6 DRACH motifs, but each motif is ligated into a separate homopolymer, i.e., one type of oligo per strand. We illustrate the difference between the two different approaches in Figure 1a.

By generating two types of data sets, we add more complexity to our test and training data. The RL oligos have a length of 21 nts, whereas the SL oligos are 33 nts long, resulting in a different spacing of m⁶A sites. Furthermore, the sequence context is different between the two data sets.

In our test data, TEST1 is composed of RL heteropolymers (2 oligos in one strand) while for TEST2 SL homopolymers (1 oligo per strand) were used. This is exactly the reverse of our training dataset, which also demonstrates the robustness of the method itself regardless of the synthesis method.

7. Please describe the results for each figure within the figure legends in detail .

Response: Following the reviewer's suggestion, we have now substantially expanded the descriptions in the figure legends.

Reviewer #3 (Remarks to the Author):

In the manuscript, Chan et al describe a new method to detect m6A from nanopore direct RNA-Seq data for individual RNA reads. To be able to train the model, the authors generated new data such that the modification status of each read is known. The authors used synthetic sequences that contain a single m6A site (or non m6A site) surrounded by 21-33 bp unmodified RNA sequence from the human transcriptome corresponding to known m6A positions. The data contains 6 different DRACH motifs, capturing 80% of human DRACH sites, and the different sequences are combined using two strategies. The computational method that is then introduced uses these data to train a logistic regression classifier that uses the features from the RODAN basecaller to predict m6A for individual reads. The method is evaluated using synthetic sequences and human RNA, and compared against other methods that are able to perform single molecule m6a prediction.

The results show that the proposed method has a high accuracy and a high similarity with GLORI-Seq results, suggesting that the proposed approach to train a single molecule m6a classifier using the newly generated training data provides an advantage.

The training data that is described in this study is different from what others have used and address an important point by enabling the training of a single molecule classifier. The manuscript is short, and overall, well written and sufficiently clear. I have some concerns as outlined below:

Major comments:

(1) The data, the training procedure, and the evaluation procedure are not sufficiently described

From the manuscript it was not sufficiently clear how the training data is generated, how many samples were generated, and how the data was used for model training and evaluation:

- The data is currently not available (or at least I could not find any link in the manuscript). Please make the data available

Response: We have uploaded the oligo and HEK293 WT sequencing data to zenodo. For clarity, there is now a section in the supplement called “Links for reviewers”.

- It is not clear how modified and unmodified data are generated. Did the authors generate different libraries using the unmodified and modified oligos? How many replicates were generated? From the current description it is unclear how the authors obtain data with labels for each read as the description is currently missing.

Response: We have added the section “Composition of ligation reactions and sequencing libraries” in the methods section to describe in more detail how the different libraries are generated and sequenced. In short, for the training dataset, the UNM and MOD libraries are generated and sequenced separately.

The testing datasets (TEST1 and TEST2) are generated differently from the training libraries, in the sense that “each strand of RNA contains both MOD and UMD oligos in unknown order.” (paragraph 5 of main text)

In the methods section “Training on synthetic molecules”, we added the following: “Reference oligo sequences are mapped locally to each read and then chained together to infer the ligated sequence in each RNA strand.”

The number of replicates depends on the number of samples we collect from the first run. Where more samples are needed, the experiment is repeated and the data is combined with the previous runs. As we analyze Nanopore reads on single-read level and work with synthetic RNA oligos, we don’t require “replicates” in the classical sense. For the HEK293 WT we provide two replicates on the level of isoform prediction in Figure S5c. We now list all sequencing runs in Supplementary Table T3.

In addition, we have added more descriptions to Figure 1a to clarify the two ligation strategies.

- is one model learned for all kmers, or do the authors train a separate model for each kmer?

Response: We have expanded the section “mAFiA module” in the Supplement to explain the procedure in more detail. Each binary classifier is specific to a 5mer, and is simply a 768-dimensional weight vector w_i and a scalar b_i that performs a logistic regression:

$$P(m^6 A) = \sigma(w_i \cdot x + b_i)$$

where $i \in \{1, \dots, 6\}$. So in total, there are 6 row vectors and 6 scalars. They can be combined into a 6x768 matrix W and a 6-dimensional column vector B. But for the current release, they are kept separate.

- how does the model perform when reads from different kmers are analysed together (i.e. one precision recall curve for all kmers)? Such an analysis could help to test if some kmers will be

ranked lower than others, which would impact results for example on human data where all kmers are analysed together

Response: The methods section “Composition of ligation reactions and sequencing libraries” contains details on how the ligation products are sequenced. The random ligation (RL) RNAs are sequenced in pools that contain 2-4 kmers, while the splint ligation (SL) RNAs contain 3 kmers in each single strand. As mentioned now in “Training on synthetic molecules”, “Reference oligo sequences are mapped locally to each read and then chained together to infer the ligated sequence in each read. Afterwards, each central A nucleotide in an oligo is classified into one of the six target DRACH motifs.” So the starting point of our analysis already has multiple kmers mixed together not just *in silico*, but *in vitro*, either in the same read or in the same pool.

As our method performs predictions on single positions of a read, mixing the reads from different kmers in-silico is equivalent to taking a weighted average of each kmer’s precision-recall curve. In reporting the overall AUC of each combination of training sets (Figure S3b), we use equal weight for each motif, since their sample size in the synthetic data does not necessarily represent their real abundance in any biological sample. The total number of samples collected for each motif are tabulated in Tables T4 and T5.

Our TEST1 (Figures 1e-g) mixes 2 different kmers - 1 modified and 1 unmodified - in a single read ($\{UGm^{\circ}ACU, GGACC\}$ in TEST1a; $\{UGACU, GGm^{\circ}ACC\}$ in TEST1b). While the sample sizes obtained for the two kmers are slightly unbalanced, we observe a combined precision-recall curve (Figure 1g) that is comparable to what was achieved in the validation PRCs in Figure 1d.

In human data, the location of each kmer on a read is well-defined, and, as mentioned before, our method works at single-nucleotide resolution. Hence, we do not expect the existence of multiple kmers on a read to influence the accuracy of mAFiA.

- The method is evaluated on a balanced data set it seems. How are the two classes defined? Is the data filtered in any way to obtain balanced data?

Response: Features collected from the unmodified (UNM) libraries are given the label 0, whereas those from the modified (MOD) libraries are given the label 1. They are categorized by the surrounding 5mer in which the A resides. The motif label $i \in \{1, \dots, 6\}$ is not an explicit label for training or testing, but only informs us on which binary classifier to use. This is now explained in the methods section “mAFiA module”.

When one class contains more samples than the other, we down-sample the larger class to obtain balanced data. This detail is now mentioned in the supplementary section “Training on synthetic molecules”.

- the oligo sequences contain several positions with A kmers that are not at the center, and which could be used as negative data points (for training or just for testing). Does the model accurately discriminate the unmodified (non center) kmer/positions at reads from the modified (center) kmer/positions at reads?

Response: While there are other unmodified A positions within the sequence, they are not matched by modified $m^{\circ}A$ positions with the same kmer. Hence we have not attempted to use them for training or testing.

(2) Comparison with other methods for single molecule accuracy

- The authors evaluate their own model using Precision Recall curves. However, the comparison with other methods is only done using the AB|BA design. The authors should also compare other methods using Precision-Recall curves on the same data set.

Response: Following the reviewer's suggestion, we now compare with other methods in TEST1 and TEST2 using only precision-recall-curves.

- I did not fully understand the AB|BA AB-BA design, or why this was used to evaluate single molecule predictions. I assume this is related to the design of the sequences which might be either including all modified center positions, or all unmodified center positions, so the authors designed this experiment to have reads with both modified and unmodified positions while still having approximate labels. This experiment could be better motivated and explained.

Response: We admit that the design of this experiment was previously not well-explained given the limited space of the short letter. Our primary motivation was to show that mAFiA is able to detect different underlying m⁶A patterns despite identical repetition of the same sequence in one read. The AB-BA design was meant to be a random baseline in which such an underlying pattern is erased, as opposed to AB|BA where the modification pattern is regular.

In the revised version of the manuscript, we now present only the AB|BA part (Figures 1h-j), for which read-level ground truth can at least be inferred from the detection results. Furthermore, we now compare mAFiA with other methods using only precision and recall as metric (PRC).

- It's unclear why the AB-BA/AB|BA experiment is used to compare mAFiA with other methods, as the evaluation depends on selecting a threshold for the read probability, whereas the precision recall would not require to select a threshold (see comment above)

Response: As suggested by the reviewer, with the PR-curve, we do not need to select a threshold anymore.

- The authors choose a threshold based on the range of 0.1 to 0.9. This range will cover most probability values for a classifier learned on balanced data, where the expected probability might be 0.5 (as in Figure 1c). However, it's not clear if that range covers CHEUI and m⁶anet as well. For completeness, the authors should evaluate values between 0 and 0.1 and 0.9 and 1 as well.

Response: As with the previous reply, the PR-curve now scans all possible thresholds from p=0.00 to 1.00.

- Another control is to compare the modification ratios that are calculated from m⁶anet and CHEUI across all reads and the m⁶a probability for the position (not just the single molecule predictions) with mAFiA for the AB-BA/AB|BA experiment.

Response: When all reads are aggregated together to produce site-level predictions, we observe stoichiometries of approximately 50% across each 13-nt cycle (Figure 1h, bottom). This agrees with theory, but we do not expect to see significant differences across the different methods, since the underlying m⁶A pattern (peak and trough at 26nt intervals) only manifests itself on a single-read basis. When the reads are mixed together, the m⁶A pattern gets completely washed out.

(3) prediction of m6A sites (not single molecule predictions)

- While the ability to predict single molecules with mAFIA using the training data design is clearly a novelty, ultimately the ability to detect which positions are modified will still be important for most use cases. Currently this aspect is not evaluated. How well can the authors identify m6A sites? This evaluation could be done with the synthetic sequences by combining all data from all motifs (which will be a balanced data set), and in comparison with the GLORI-Seq results where many more unmodified sites exist and where the kmers have different frequencies, resulting in an unbalanced but more realistic scenario. How is the precision-recall in these 2 scenarios?

Response: This comment contains multiple components, and we will attempt to answer each one separately:

For oligo RNAs, modification sites on the ensemble level are ill-defined, since each strand contains a random combination of oligo fragments and the reads do not align to a single reference. The only exception are the homopolymers, in which each strand contains only one type of oligo. However, this is precisely the RL samples on which we trained our classifiers. From the methodological point of view, we cannot test other methods on our own training set.

As for human mRNA, measurements by GLORI come in the form of site-level stoichiometry S in the range 10-100%. These numbers do not explicitly translate into modified / unmodified status of each site. To do so would involve introducing a threshold. Since single-molecule methods (mAFiA, m6Anet, and CHEUI) can all predict site-level stoichiometry, we believe that it is more natural to directly calculate the correlation between S predicted by different methods, as presented in Figures 2d, e and S4e, f.

In terms of identifying modified sites out of many unmodified locations, miCLIP does provide a set of ground-truth sites, against which one can calculate a precision-recall curve. We therefore tested mAFiA, m6Anet, and CHEUI on the same HEK293 dataset, and limited the list of miCLIP sites to those where the data has sufficient coverage (min. 50 reads). The results are shown in supplementary Figure S4d (bottom). For this comparison, all computational methods show less-than-ideal overlap with miCLIP, although mAFiA still recalls more sites at the same level of precision. We believe that miCLIP on its own also misses/mis-identifies many sites, which leads to the relatively low precision-recall performance of all Nanopore-based computational methods.

- Currently it is not clear if the model can handle unbalanced data with a much larger number of unmodified positions. These positions seem to be excluded from the comparison with GLORI-Seq (p -value < 0.001). How does mAFiA stoichiometry results look for positions where mAFiA is confident (>50 reads) but GLORI-Seq is not confident?

Response: Below we plot the comparison for GLORI-Seq p -value ≥ 0.001 and mAFiA coverage ≥ 50 :

There are very few overlapping sites in this regime (18 in total). The authors of GLORI-Seq published their data with maximum P-value 0.004, and stoichiometry S above 10%. We have investigated the GLORI sites with P-value in the range (0.001, 0.004), and observed that their site stoichiometries are in the range 10-25% (bottom panel below):

As shown above, many sites in our original comparison with GLORI (P-value < 0.001, top panel above) also have low stoichiometry, and we decided that the former P-value threshold at 0.001 is unnecessary after all.

In the revised figures, we compare against all possible GLORI sites with coverage above 50 in our dataset. The overall correlation is essentially the same (before 0.85, now 0.86).

We would like to add that most of the site-level predictions by mAFiA are in the low S range (Figure S4a). The problem of comparing these sites is that most other experimental methods look for highly modified sites only. GLORI does publish site stoichiometries down to $S=10\%$, but their confidence there is also not high.

- CHEUI and m6anet make site level predictions, how do the site level predictions compare against the predictions from mAFiA for individual positions? This comparison can again be done using the precision-recall to avoid having to select a threshold

Response: We would like to emphasize that, just like CHEUI and m6Anet, mAFiA makes both read-level (in the form of positional $P(m^6A)$ on single reads) as well as site-level predictions (in the form of site stoichiometry S). We have added the following sentences to paragraph 7 of the main text to stress the difference between them:

“To aggregate the distribution of read-level $P(m^6A)$ aligned to a column into site-level modification ratio, we define the site stoichiometry S as the fraction of modified nucleotides, ie, nts with $P(m^6A) \geq 0.5$, aligned to that site (see also definition of S in supplement).”

There is now also a dedicated section in the methods section “Definition of S” that explains the relationship between the two quantities.

Related to a previous comment, we can only compare computational methods based on a set of experimental ground-truth. On the site-level, we have compared against GLORI-Seq using correlation, and against miCLIP using 0/1 site labels. For the latter comparison, the predictions of all 3 methods on HEK293 are compared with threshold-free PR curves (Figure S4d bottom). mAFiA yields higher recall at the same precision than the other two methods.

(4) Identification of m6a in human and non-human data/sequence context

- The model is trained on data that captures the complexity of human m6A sites. However, other species such as yeast or arabidopsis, or synthetic data will have a different m6A motifs, and it is not clear how well the model will work on such data. How well does their method identify m6A in non-human sequences? If the model does not identify m6A in non-human RNA, or if it only captures a small percentage of sites due to the specific motif selection, then this should be clearly described to avoid users applying it to data that is not supported by this model.

Response: We have added this paragraph to the main text:

“While mAFiA is optimized for the most common m^6A patterns in the human transcriptome, we evaluated its applicability also in a wider context. Testing on samples of *Arabidopsis thaliana* dRNA data shows good correspondence to previously published miCLIP measurements⁹ (Figure 2g). Out of 522 high-modification sites predicted (sites with $S_{mAFiA} \geq 50\%$), 372 (71%) coincide with a miCLIP peak within 5 nts. The agreement rises to 82% if we consider only the more confident sites with $S_{mAFiA} \geq 80\%$. A site-by-site comparison of the predicted m^6A profiles between the *col0* (wildtype) and *vir1* (mutant) strains shows a transcriptome-wide down-regulation in the otherwise highly modified sites (Figures 2h and S6). We note that the overall coverage of m^6A sites in a specific species can be further improved with bespoke training data, although the primary use case of mAFiA remains mammalian RNAs.”

Minor:

-figure 1c: How was the data normalised? How do these figures look without normalisation?

Response: The histograms were originally normalized such that all bins in each histogram sum to a total probability of 1. In the revised Figure 1c, we show the absolute numbers instead, with the numbers listed in the legend.

- Figures 1f and 1h are unclear. What is the x-axis? Are all positions predicted to have $P(m^6A) = 0.8$?

Response: The original figures were intended to show the spacing between the predicted $P(m^6A)$ on a read-by-read basis. We do not make predictions on the nucleotides between them. In the revised figure, we show the results on an IGV snapshot instead for TEST2 (Figure 1h) and as violin plots (Figure 1f, i).

- Figure 1a, the schematic does not sufficiently explain the design. What are the colours? Which sequences have m6a, which don't have m6a? A clearer illustration would be very helpful (see also point 1)

Response: We have added more details to Figure 1a and its accompanying caption to better explain the generation of our training and validation data based on enzymatic ligation of synthetic RNA oligos.

- "Unlike other single-molecule methods, mAFIA does not require additional post-processing such as nanopolish7, and can be integrated into an existing basecaller without altering the latter's accuracy." I assume that most nanopore users by default get the basecalled sequences with the nanopore basecaller. The mAFIA method requires users to run a different basecaller (RODAN), which also requires resources in addition to the nanopore basecalling then. If the authors want to keep the claim that their method has minimal computational overhead, then they should compare the resource requirements for RODAN vs Nanopolish, or the authors use features from the default output of nanopore basecalling. Alternatively, I would suggest to specify these claims to make clear that this is only true when the RODAN basecaller is used.

Response: In terms of run-time and accuracy, RODAN is actually comparable to ONT guppy. Indeed, all mAFiA results were obtained without running guppy. However, we understand that most users would prefer using the official software. So, we have removed our claim about the overhead from the main text.

The specifics of run-time and computational requirements are detailed in the Supplementary section "Computational requirement and run-time".

- Related to this, since the RODAN basecaller seems to be essential to the manuscript, I would recommend to mention RODAN in figure 1b, in the main text, and to cite the paper in the main text references to fully acknowledge the method.

Response: We now reference RODAN prominently in the main text (paragraph 3), Figure 1b, as well as in the corresponding caption.

REVIEWERS' COMMENTS

Reviewer #2 (Remarks to the Author):

The revised manuscript has addressed all comments raised satisfactorily during the initial review. The changes made significantly contribute to the clarity and scientific rigor of the work and we believe that the manuscript now aligns well with the standards expected for publication in Nature Communications.

Reviewer #3 (Remarks to the Author):

In the revised manuscript, the authors have made changes to improve the description and presentation of the method and results. I don't have further comments.

Reviewer #3 (Remarks on code availability):

The code documentation seems sufficient. The method is not available through a standard package manager, and running the method requires several steps, therefore I expect this to be aimed at a more specialised audience.